# Changes in BNP levels from discharge to 6-month visit predict subsequent outcomes in patients with acute heart failure

Masayuki Shiba[1], Takao Kato[1]*, Takeshi Morimoto[2], Hidenori Yaku[3], Yasutaka Inuzuka[4], Yodo Tamaki[5], Neiko Ozasa[1], Yuta Seko[1], Erika Yamamoto[1], Yusuke Yoshikawa[1], Takeshi Kitai[6], Yugo Yamashita[1], Moritake Iguchi[7], Kazuya Nagao[8], Yuichi Kawase[9], Takashi Morinaga[10], Mamoru Toyofuku[11], Yutaka Furukawa[12], Kenji Ando[10], Kazushige Kadota[9], Yukihito Sato[13], Yasuaki Nakagawa[1], Koichiro Kuwahara[14], Takeshi Kimura[1]

1 Department of Cardiovascular Medicine, Kyoto University Graduate School of Medicine, Kyoto, Japan, 2 Department of Clinical Epidemiology, Hyogo College of Medicine, Nishinomiya, Japan, 3 Department of Cardiology, Mitsubishi Kyoto Hospital, Kyoto, Japan, 4 Cardiovascular Medicine, Shiga General Hospital, Moriyama, Japan, 5 Division of Cardiology, Tenri Hospital, Tenri, Japan, 6 Division of Heart Failure, National Cerebral and Cardiovascular Center, Suita, Japan, 7 Department of Cardiology, National Hospital Organization Kyoto Medical Center, Kyoto, Japan, 8 Department of Cardiology, Osaka Red Cross Hospital, Osaka, Japan, 9 Department of Cardiology, Kurashiki Central Hospital, Kurashiki, Japan, 10 Department of Cardiology, Kokura Memorial Hospital, Kokura, Japan, 11 Department of Cardiology, Japanese Red Cross Wakayama Medical Center, Wakayama, Japan, 12 Department of Cardiovascular Medicine, Kobe City Medical Center General Hospital, Kobe, Japan, 13 Department of Cardiology, Hyogo Prefectural Amagasaki General Medical Center, Amagasaki, Japan, 14 Department of Cardiovascular Medicine, Shinshu University Graduate School of Medicine, Nagano, Japan

* tkato75@kuhp.kyoto-u.ac.jp

**Data Availability Statement:** All relevant data are within the paper and its Supporting information files.

## Abstract

### Background

This study aimed to investigate the association between changes in brain natriuretic peptide (BNP) from discharge to 6-month visit and subsequent clinical outcomes in patients with acute heart failure (AHF).

### Methods

Among 1246 patients enrolled in the prospective longitudinal follow-up study nested from the Kyoto Congestive Heart Failure registry, this study population included 446 patients with available paired BNP data at discharge and 6-month index visit. This study population was classified into 3 groups by percent change in BNP from discharge to 6-month visit; the low tertile ($\leq$-44%, N = 149), the middle tertile (>-44% and $\leq$22%, N = 149) and the high tertile (>22%, N = 148).

### Findings

The cumulative 180-day incidence after the index visit of the primary outcome measure (a composite endpoint of all-cause death or hospitalization for HF) was significantly higher in the high and middle tertiles than in the low tertile (26.8% and 14.4% versus 6.9%, log-rank P<0.0001). The adjusted excess risk of the high tertile relative to the low tertile remained

**Funding:** This work was supported by the Japan Agency for Medical Research and Development [18059186] (T.K, K.K, and N.O). https://www.amed.go.jp/en/ The funders had no role in study design, data collection and analysis, decision to publish, or preparation of the manuscript.

**Competing interests:** The authors have declared that no competing interests exist.

significant for the primary outcome measure (hazard ratio: 3.43, 95% confidence interval: 1.51–8.46, P = 0.003).

## Conclusions

Percent change in BNP was associated with a subsequent risk for a composite of all-cause death and hospitalization for HF after adjustment of the absolute BNP values, suggesting that observing the change in BNP levels, in addition to absolute BNP levels themselves, helps us to manage patient with HF.

## Introduction

Natriuretic peptide is the powerful biomarker for diagnosis of acute and chronic heart failure (HF) [1, 2]. Brain natriuretic peptide (BNP) measurement is strongly recommended and widely applied in daily clinical HF management [3, 4]. A high value of BNP level is an important parameter for worsening HF [5]. However, absolute BNP values in a compensated condition varied in each patient due to the underlying cardiac disease, the extent of ventricular hypertrophy, pre-loads and after-loads as well as many non-cardiac factors such as age, renal failure and obesity [6, 7].

In addition to baseline BNP levels, changes in BNP are also important in the management for HF [4, 8]. Previous studies showed that a change in BNP from admission for acute HF (AHF) to follow-up after discharge was associated with clinical events in HF patients [8]. However, in clinical practice, HF management was supported by BNP levels at follow-up, compared with BNP levels at discharge in a steady condition and the prognostic value of the changes in BNP levels from discharge to follow-up remains to be elucidated. Therefore, in the present study, we investigated the association between the changes in BNP levels and subsequent clinical outcomes in patients with AHF.

## Materials and methods

### Patient population

In the Kyoto Congestive Heart Failure (KCHF) registry, we enrolled consecutive 4,056 patients who were hospitalized for AHF as index hospitalization between 1 October 2014 and 31 March 2016. Identifiable patient records were anonymized before analysis. The detailed description of rationale, design and enrollment of the KCHF registry have been previously showed [9, 10]. In the prospective longitudinal follow-up study parallel with the main KCHF study, we enrolled 1,246 patients who were to have a visit at 6 +/- one month after excluding 271 patients who died during index hospitalization and 2,539 patients corresponding to exclusion criteria [10]. The design and exclusion criteria for the prospective longitudinal follow-up study has been specifically described in our previous reports [10]. After excluding 99 patients who were lost to follow within 6 months after the index hospitalization or after a 6-month visit, 23 patients who died within 6 months after the index hospitalization, and 678 patients with missing BNP data at discharge and/or at 6-month visit (S1 Table), the present study population consisted of 446 patients with paired data available for serum BNP (Figs 1 and 2). This study population was classified into the 3 groups by tertiles of percent change in BNP levels from discharge to 6-month visit.

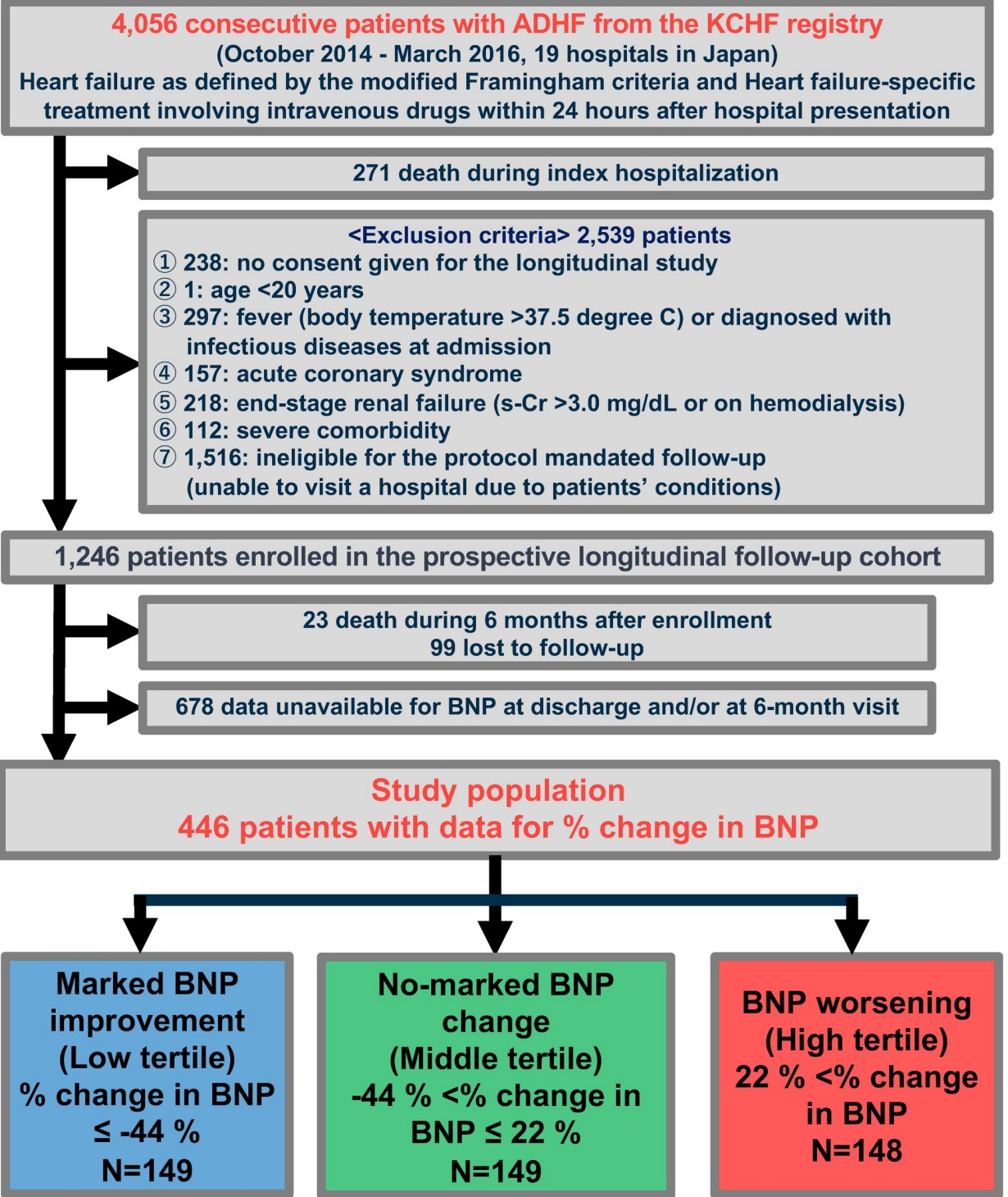

**Fig 1. Study flowchart.** AHF, acute heart failure; KCHF, Kyoto Congestive Heart Failure; s-Cr, serum creatinine; BNP, brain natriuretic peptide.

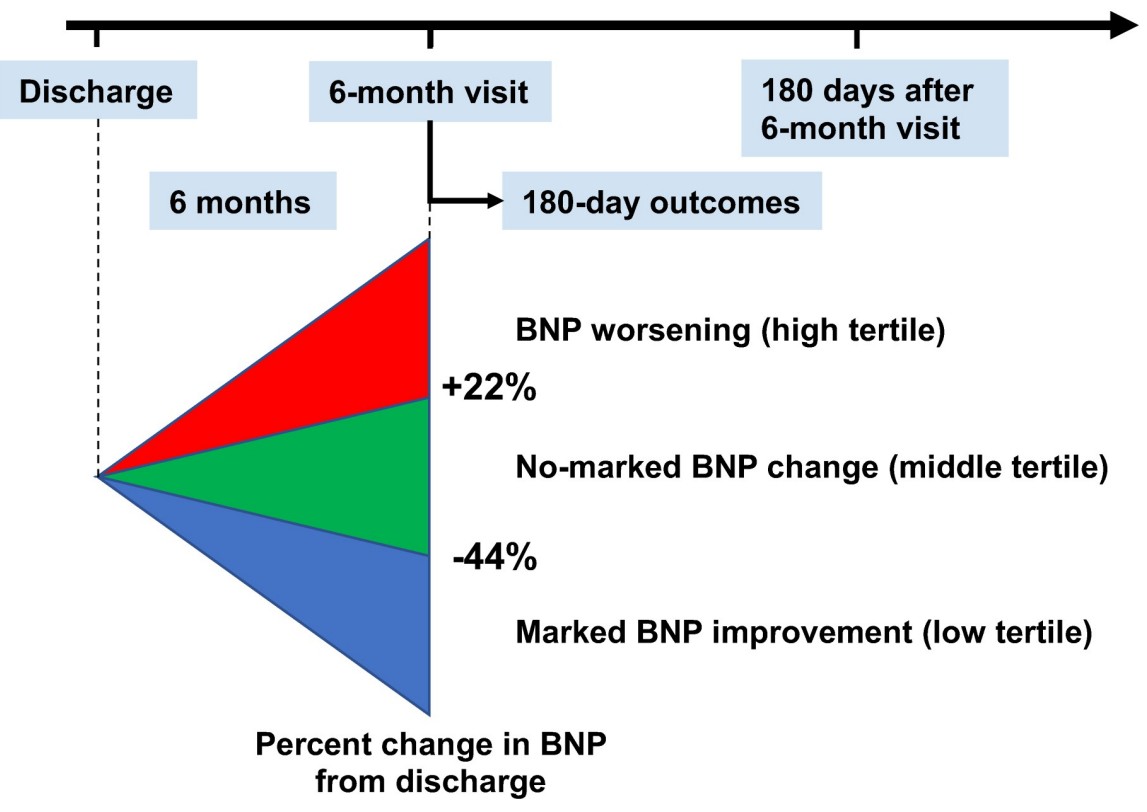

**Fig 2. Scheme of the present analysis.** BNP, brain natriuretic peptide.

### Ethics

The investigation conformed with the principles outlined in the Declaration of Helsinki. The study protocol was approved by the ethical committees of the Kyoto University Hospital (local identifier: E2311) and each participating hospital. Written informed consent was obtained from patients enrolled in the longitudinal prospective cohort study.

### Outcomes

The date of the 6-month visit was considered as time zero for evaluating the clinical events censored at 210 days after the 6-month visit in this study (Fig 2). The primary outcome measure in this study was defined as a composite of all-cause death or hospitalization for HF [10]. The secondary outcome measure was all-cause death and hospitalization for HF, respectively.

### Definitions

AHF is defined as de novo HF or worsening signs and symptoms of HF [11]. A change in BNP levels was calculated as follows: (BNP level at 6-month visit)—(BNP level at discharge). Percent change in BNP level was calculated by dividing the change in BNP by the BNP level at discharge and multiplying the result by one hundred to make it a percentage. The detailed definitions of baseline patient characteristics were previously described [9, 10]. HF was divided into HF with reduced left ventricular ejection fraction (LVEF) (<40%) (HFrEF) and HF with non-reduced LVEF (≥40%) (non-HFrEF), based on LVEF at the 6-month visit. Atrial arrythmias

including atrial fibrillation and flutter were counted base on medical history and their events during index hospitalization and electrocardiography at 6-month visit.

## Statistical analysis

Continuous variables were expressed as mean and standard deviation or median with interquartile range (IQR) and categorical variables are expressed as counts and percentages. Differences among the 3 groups were evaluated by means of the one-way analysis of variance, the Kruskal-Wallis test or the chi-square test, as appropriate. A paired t test was used for continuous variables and Sign test was used for binary variables to compare those at discharge and those at 6-month visit. Cumulative incidences were calculated by means of the Kaplan–Meier analysis and the among-groups differences are tested by means of the log-rank test.

We set the low tertile of percent change in BNP as reference and evaluated the adjusted risks of high tertile versus low tertile, and middle tertile versus low tertile for the primary and secondary outcome measures. The Cox proportional hazards regression models were utilized to assess the association between percent change in BNP levels and the clinical events after adjusting for 10 clinically relevant risk variables: age $\geq$80 years, sex, LVEF <40% by echocardiography, BNP levels at discharge, eGFR <30ml/min/1.73m$^2$, albumin <3.0 g/dL and medications at 6-month visit (diuretics, angiotensin converting-enzyme inhibitor [ACE-I] or angiotensin-receptor blocker [ARB], β-blocker, and mineralocorticoid receptor antagonist [MRA]). As a sensitivity analysis, we included age, LVEF, BNP at discharge, eGFR and albumin as a continuous variable in the adjusted model in patients with available data. The results were expressed as the hazard ratios (HRs) and their 95% CIs. Post-hoc subgroup analyses were performed in the 5 clinically relevant subgroups including the tertiles of BNP levels at 6-month visit ($\leq$120 ng/L, >120 ng/L and $\leq$295 ng/L, and >295 ng/L), atrial arrythmias, LVEF <40%, use of ACE-I or ARB, and use of β-blocker at 6-month visit. Effects of percent change in BNP levels-by-subgroup interactions were evaluated by means of the Cox proportional hazards regression model. In additional subpopulation analyses, this study population (446 patients) was classified into 3 subpopulations according to BNP level at discharge; low tertile ($\leq$155 ng/L, N = 150), middle tertile (>155ng/L and $\leq$350 ng/L, N = 147), and high tertile (>350 ng/L, N = 149). We compared cumulative incidences of the tertiles of percent change in BNP by means of the log-rank test in each BNP level at discharge. Statistical analyses were performed using JMP pro software, version 14 (SAS Corp., Cary, NC, USA). A two-tailed P value <0.05 was considered as statistically significant in all analyses.

## Results

### Clinical characteristics, laboratory test results, and medications at 6-month visit

The study population was classified into the 3 groups by tertiles of percent change in BNP levels; the low tertile, the marked BNP improvement group ($\leq$-44%, N = 149), the middle tertile, the no-marked BNP change group (>-44% and $\leq$22%, N = 149) and the high tertile, the BNP worsening group (>22%, N = 148) (S1 Fig). Baseline characteristics at 6-month visit were significantly different across the 3 groups (Table 1). Compared with the marked BNP improvement group, the BNP worsening and no-marked BNP change groups were older and had higher prevalence of woman, a history of atrial arrhythmia, and myocardial infarction (Table 1). Compared with the marked BNP improvement group, the BNP worsening and no-marked BNP change groups had lower serum albumin, hemoglobin and eGFR and had a

**Table 1. Patient characteristics at 6-month visit.**

| Variable | Marked BNP improvement (low tertile) (N = 149) | No-marked BNP change (middle tertile) (N = 149) | BNP worsening (high tertile) (N = 148) | P value | N of patients analyzed |
|---|---|---|---|---|---|
| **Clinical characteristics** | | | | | |
| Age (years) | 69.5 ± 13.8 | 76.8 ±11.7 | 78.3 ± 9.4 | <0.0001 | 446 |
| Age ≥80 years [a] | 39 (26%) | 77 (52%) | 79 (53%) | <0.0001 | 446 |
| Women [a] | 56 (38%) | 69 (46%) | 84 (57%) | 0.004 | 446 |
| BMI (kg/m$^2$) | 23.2 ± 5.3 | 22.7 ± 4.9 | 22.9 ± 4.7 | 0.48 | 353 |
| BMI ≤22 kg/m$^2$ | 49 (40%) | 61 (52%) | 56 (49%) | 0.19 | 353 |
| **Etiology** | | | | | |
| Coronary artery disease | 34 (23%) | 36 (24%) | 38 (26%) | 0.85 | 446 |
| Hypertensive heart disease | 41 (28%) | 44 (30%) | 47 (32%) | 0.73 | 446 |
| Cardiomyopathy | 50 (34%) | 32 (21%) | 24 (16%) | 0.002 | 446 |
| Valvular heart disease | 17 (11%) | 26 (17%) | 25 (17%) | 0.26 | 446 |
| Arrythmia | 4 (2.7%) | 8 (5.4%) | 10 (6.8%) | 0.23 | 446 |
| Other diseases | 3 (2.0%) | 3 (2.0%) | 4 (2.7%) | 0.90 | 446 |
| **Medical history** | | | | | |
| AF or AFL | 69 (46%) | 102 (68%) | 98 (66%) | 0.0001 | 446 |
| Hypertension | 103 (69%) | 110 (74%) | 114 (77%) | 0.30 | 446 |
| Diabetes | 50 (34%) | 54 (36%) | 62 (42%) | 0.32 | 446 |
| Dyslipidemia | 57 (38%) | 62 (42%) | 54 (36%) | 0.65 | 446 |
| Previous myocardial infarction | 23 (15%) | 34 (23%) | 44 (30%) | 0.01 | 446 |
| Previous ischemic stroke or ICH | 21 (14%) | 21 (14%) | 20 (14%) | 0.99 | 446 |
| Chronic lung disease | 20 (13%) | 14 (9.4%) | 17 (11%) | 0.55 | 446 |
| **Vital signs at 6-month visit after discharge** | | | | | |
| Systolic BP (mmHg) | 124.7 ± 19.8 | 119.5 ± 21.0 | 122.7 ± 21.5 | 0.04 | 387 |
| HR (bpm) | 74.0 ± 12.7 | 73.5 ± 14.2 | 77.6 ± 16.2 | 0.16 | 384 |
| **BNP values at discharge and 6-month visit** | | | | | |
| BNP at discharge (ng/L) [a] | 318 (157–537) | 239 (150–447) | 160 (87.3–301) | <0.0001 | 446 |
| BNP at 6-month visit (ng/L) | 73.1 (28.6–149) | 218 (128–380) | 370 (185–727) | <0.0001 | 446 |
| Change in BNP (ng/L) | -212 (-396- -97.7) | -27.7 (-71.1–5) | 153 (72.1–361) | <0.0001 | 446 |
| % change in BNP (%) | -72.3 (-84.2- -59.3) | -12.7 (-31.1–2.2) | 92.0 (44.6–197) | <0.0001 | 446 |
| **Tests at 6-month visit after discharge** | | | | | |
| eGFR (mL/min/1.73m$^2$) | 51.0 ± 22.6 | 41.2 ± 16.1 | 43.6 ± 20.3 | 0.0002 | 445 |
| eGFR <30 mL/min/1.73m$^2$ [a] | 24 (16%) | 40 (27%) | 44 (30%) | 0.01 | 445 |
| Albumin (g/dL) | 4.1 ± 0.50 | 3.9 ± 0.56 | 3.8 ± 0.53 | <0.0001 | 427 |
| Albumin <3 g/dL [a] | 3 (2.1%) | 5 (3.5%) | 9 (6.3%) | 0.19 | 427 |
| Sodium (mEq/L) | 139.1 ± 3.5 | 140.1 ± 3.1 | 140.2± 3.1 | 0.01 | 445 |
| Hemoglobin (g/dL) | 12.6 ± 2.0 | 11.9 ± 1.9 | 11.3 ± 2.2 | <0.0001 | 440 |
| **Medications at 6-month visit after discharge** | | | | | |
| ACE-I or ARB [a] | 80 (67%) | 73 (61%) | 73 (62%) | 0.56 | 356 |
| MRA [a] | 58 (49%) | 51 (42%) | 53 (45%) | 0.55 | 356 |
| β-blocker [a] | 102 (85%) | 93 (78%) | 84 (72%) | 0.04 | 357 |

(*Continued*)

**Table 1.** (Continued)

| Variable | Marked BNP improvement (low tertile) (N = 149) | No-marked BNP change (middle tertile) (N = 149) | BNP worsening (high tertile) (N = 148) | P value | N of patients analyzed |
|---|---|---|---|---|---|
| Diuretics [a] | 94 (79%) | 102 (84%) | 107 (91%) | 0.04 | 358 |

Categorical variables are presented as number (%), and continuous variables are presented as mean ± SD or median (interquartile range).

Diuretics included loop diuretic, thiazide and tolvaptan.

[a] Risk-adjusting variables selected for the Cox proportional hazards regression model: age ≥80 years, sex, BNP values at discharge as a continuous variable, eGFR <30 mL/min/1.73m$^2$, albumin <3 g/dL, ACE-I or ARB, MRA, β-blockers and diuretics, in addition to LVEF <40% at 6-month visit echocardiography in Table 2.

BMI, body mass index; AF, atrial fibrillation; AFL, atrial flutter; ICH, intracranial hemorrhage; BP, blood pressure; HR, heart rate; BNP, brain natriuretic peptide; eGFR, estimated glomerular filtration rate; ACE-I, angiotensin-converting enzyme inhibitor; ARB, angiotensin-receptor blocker; MRA, mineralocorticoid receptor antagonist; LVEF, left ventricular ejection fraction; SD, standard deviation.

lower prevalence of cardiomyopathy etiology and β-blocker use and a higher proportion of diuretics use (Table 1).

## Echocardiographic findings at discharge and at 6-month visit

The BNP worsening group had a larger left atrial diameter (LAD) and a smaller left ventricular end-diastolic dimension (LVEDD), lower left ventricular mass index (LVMI) and higher LVEF at discharge than the marked BNP improvement and no-marked BNP change groups (Table 2). At 6-month visit, the BNP worsening group had a higher LVMI and tricuspid regurgitation pressure gradient (TRPG), a greater LAD and diameter of inferior vena cava (IVC), and a higher prevalence of moderate/severe mitral regurgitation (MR) and tricuspid regurgitation (TR) than the marked BNP improvement and no-marked BNP change groups. On the other hand, there were no significant differences in LVEDD and LVEF at the 6-month visit among the 3 groups (Table 2). From discharge to 6-month visit, the BNP worsening group had a minimal increase in LVEF and a minimal decrease in LVEDD, LVMI and TRPG, and had a numerical increase in LAD, IVC diameter and the prevalence of moderate/severe TR (Table 2 and representative values in Fig 3).

## Clinical outcomes

The follow-up rate at 180-day after the 6-month visit was 97.3%. During the 180-day follow-up, 39 patients in the marked BNP improvement group, 21 patients in the no-marked BNP change group and 10 patients in the BNP worsening group encountered all-cause death or hospitalization for HF (Fig 4A and Table 3). The cumulative 180-day incidences of the primary outcome measure were significantly higher in the BNP worsening group and the no-marked BNP change group than in the marked BNP improvement group (26.8% in the BNP worsening group and 14.4% in the no-marked BNP change group versus 6.9% in the marked BNP improvement group, log-rank P <0.0001) (Fig 4A). With respect to the secondary outcome measures, the cumulative 180-day incidence of all-cause death was significantly higher in the BNP worsening group than in the no-marked BNP change group and the marked BNP improvement group (9.6%, 4.8%, and 4.1%, respectively, log-rank P = 0.04) (Fig 4B) and the cumulative 180-day incidence of hospitalization for HF was also significantly higher in BNP worsening group and the no-marked BNP change group than in the marked BNP improvement group (21.2%, 9.9%, and 2.8%, respectively, log-rank P<0.0001) (Fig 4C).

After adjusting for confounding variables, the excess risk of the BNP worsening group relative to the marked BNP improvement group remained significant for the primary outcome

PLOS ONE BNP and heart failure

**Table 2. Changes in echocardiographic parameters from discharge to 6-month visit.**

| Variable | Marked BNP improvement (low tertile) (N = 149) | | | | No-marked BNP change (middle tertile) (N = 149) | | | | BNP worsening (high tertile) (N = 148) | | | | Between-groups comparison | | |
|---|---|---|---|---|---|---|---|---|---|---|---|---|---|---|---|
| | Discharge | 6-month visit | Delta # | P value (paired) | Discharge | 6-month visit | Delta # | P value (paired) | Discharge | 6-month visit | Delta # | P value (paired) | P value (discharge) | P value (6-month visit) | P value delta |
| LVEDD (mm) | 55.1 ± 8.5 | 49.5 ± 8.5 | -5.6 ± 6.4 | <0.0001 | 52.4 ± 9.9 | 51.0 ± 10.2 | -1.7 ± 4.8 | <0.0001 | 50.6 ± 9.1 | 49.7 ± 9.1 | -0.9 ± 4.8 | 0.03 | <0.0001 | 0.70 | <0.0001 |
| LVESD (mm) | 44.0 ± 10.9 | 35.7 ± 10.1 | -8.2 ± 8.3 | <0.0001 | 40.0 ± 12.2 | 38.0 ± 12.7 | -2.4 ± 5.9 | <0.0001 | 37.4 ± 10.6 | 36.0 ± 11.2 | -1.2 ± 6.3 | 0.04 | <0.0001 | 0.66 | <0.0001 |
| IVST (mm) | 9.6 ± 2.1 | 10.1 ± 2.2 | -0.3 ± 1.6 | 0.04 | 9.5 ± 2.0 | 10.3 ± 2.4 | -0.1 ± 1.4 | 0.42 | 10.1 ± 2.2 | 10.3 ± 2.3 | -0.2 ± 1.5 | 0.16 | 0.16 | 0.10 | 0.81 |
| LVMI (g/m²) | 135.4 ± 40.0 | 108.6 ± 31.4 | -29.3 ± 32.6 | <0.0001 | 123.0 ± 35.5 | 116.2 ± 36.7 | -11.6 ± 26.9 | <0.0001 | 126.6 ± 40.6 | 124.4 ± 40.7 | -7.9 ± 27.6 | 0.004 | 0.01 | 0.005 | <0.0001 |
| LVEF (%) | 38.8 ± 16.7 | 51.8 ± 13.5 | 13.0 ± 14.3 | <0.0001 | 46.4 ± 15.7 | 49.6 ± 16.4 | 3.8 ± 11.9 | 0.0003 | 48.7 ± 15.6 | 50.3 ± 16.2 | 2.0 ± 10.1 | 0.02 | <0.0001 | 0.66 | <0.0001 |
| LVEF <40% ᵃ | 82/137 (60%) | 29/137 (21%) | -53 (-39%) | <0.0001 | 52/137 (38%) | 41/137 (30%) | -11 (-8.0%) | 0.03 | 37/139 (27%) | 39/139 (28%) | 2 (1.4%) | 0.64 | <0.0001 | 0.22 | <0.0001 |
| LAD (mm) | 43.8 ± 7.2 | 39.3 ± 8.6 | -4.2 ± 6.8 | <0.0001 | 46.5 ± 9.3 | 45.2 ± 9.1 | -1.7 ± 6.5 | 0.003 | 46.7 ± 9.6 | 47.1 ± 9.6 | 0.04 ± 6.6 | 0.94 | 0.02 | <0.0001 | <0.0001 |
| Moderate/Severe MR | 43/136 (32%) | 18/136 (13%) | -25 (-18%) | <0.0001 | 56/127 (44%) | 47/127 (37%) | -9 (-7.1%) | 0.08 | 48/130 (37%) | 48/130 (37%) | 0 (0%) | 1.0 | 0.09 | <0.0001 | 0.01 |
| Moderate/Severe TR | 40/135 (30%) | 19/135 (14%) | -21 (-16%) | 0.0003 | 38/133 (29%) | 38/133 (29%) | 0 (0%) | 1.0 | 43/135 (32%) | 49/135 (36%) | 6 (4.4%) | 0.22 | 0.90 | <0.0001 | 0.001 |
| TRPG (mmHg) | 33.0 ± 12.5 | 22.6 ± 10.3 | -9 ± 12.6 | <0.0001 | 32.3 ± 12.0 | 28.5 ± 12.0 | -3.1 ± 13.1 | <0.0001 | 33.2 ± 13.1 | 31.8 ± 13.6 | -0.1 ± 13.9 | 0.94 | 0.90 | <0.0001 | <0.0001 |
| IVC (mm) | 17.0 ± 5.4 | 13.8 ± 4.1 | -3.4 ± 5.4 | <0.0001 | 16.5 ± 4.6 | 15.5 ± 4.4 | -1.1 ± 5.3 | 0.02 | 16.0 ± 4.6 | 16.7 ± 5.1 | 0.8 ± 4.9 | 0.07 | 0.23 | <0.0001 | <0.0001 |

Categorical variables are presented as number (%), and continuous variables are presented as mean ± SD.

\# Delta is calculated for continuous variables according to the following equation: (the value at 6-month visit)−(the value at discharge) and for binary variables according to the following equation: (the numbers at 6-month visit)−(the numbers at discharge).

ᵃ Risk-adjusting variables selected for the Cox proportional hazards regression model: LVEF <40% at 6-month visit echocardiography in addition to variables in Table 1.

BNP, brain natriuretic peptide; LVEDD, left ventricular end-diastolic dimension; LVESD, left ventricular end-systolic dimension; IVST, intraventricular septum thickness; LVMI, left ventricular mass index; LVEF, left ventricular ejection fraction; LAD, left atrial diameter; MR, mitral regurgitation; TR, tricuspid regurgitation; TRPG, tricuspid regurgitant pressure gradient; IVC, inferior vena cava; SD, standard deviation; n/a, not available.

PLOS ONE | https://doi.org/10.1371/journal.pone.0263165 January 28, 2022 8 / 16

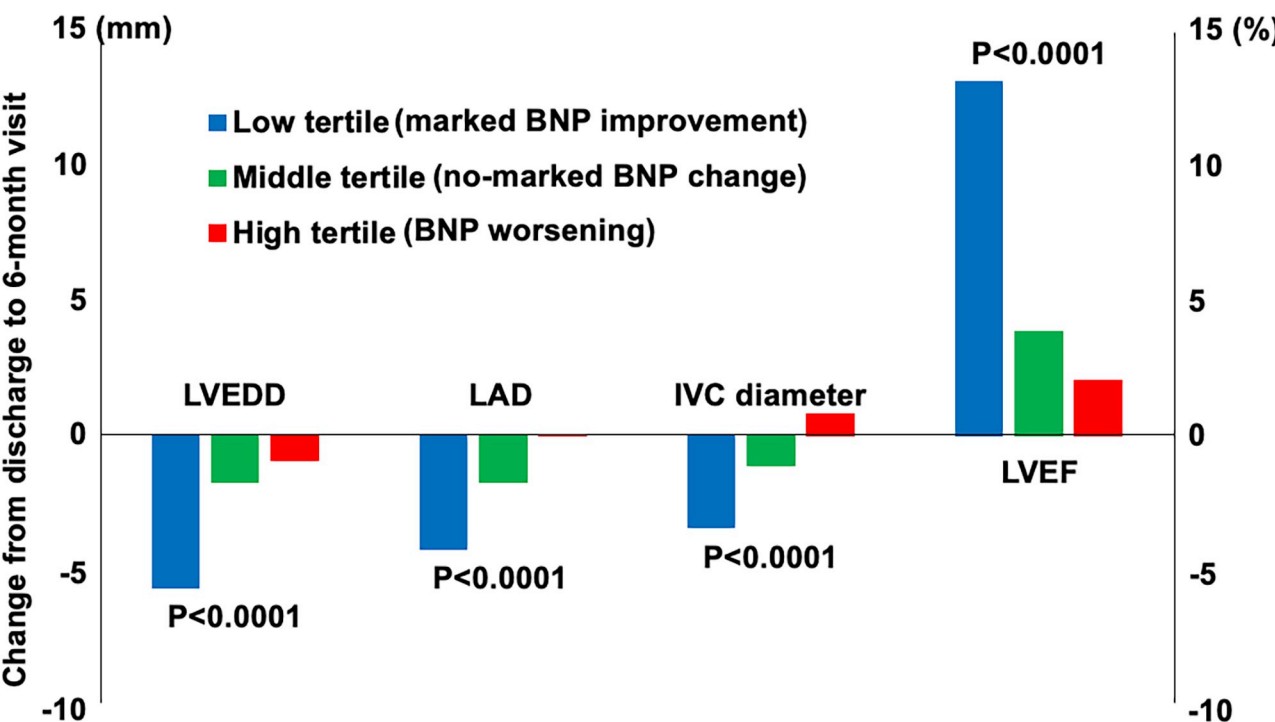

**Fig 3. Changes of echocardiographic parameters from discharge to 6-month visit.** Changes of each echocardiographic parameters are represented as mean values. LVEDD, left ventricular end-diastolic dimension; LAD, left atrial diameter; IVC, inferior vena cava; LVEF, left ventricular ejection fraction.

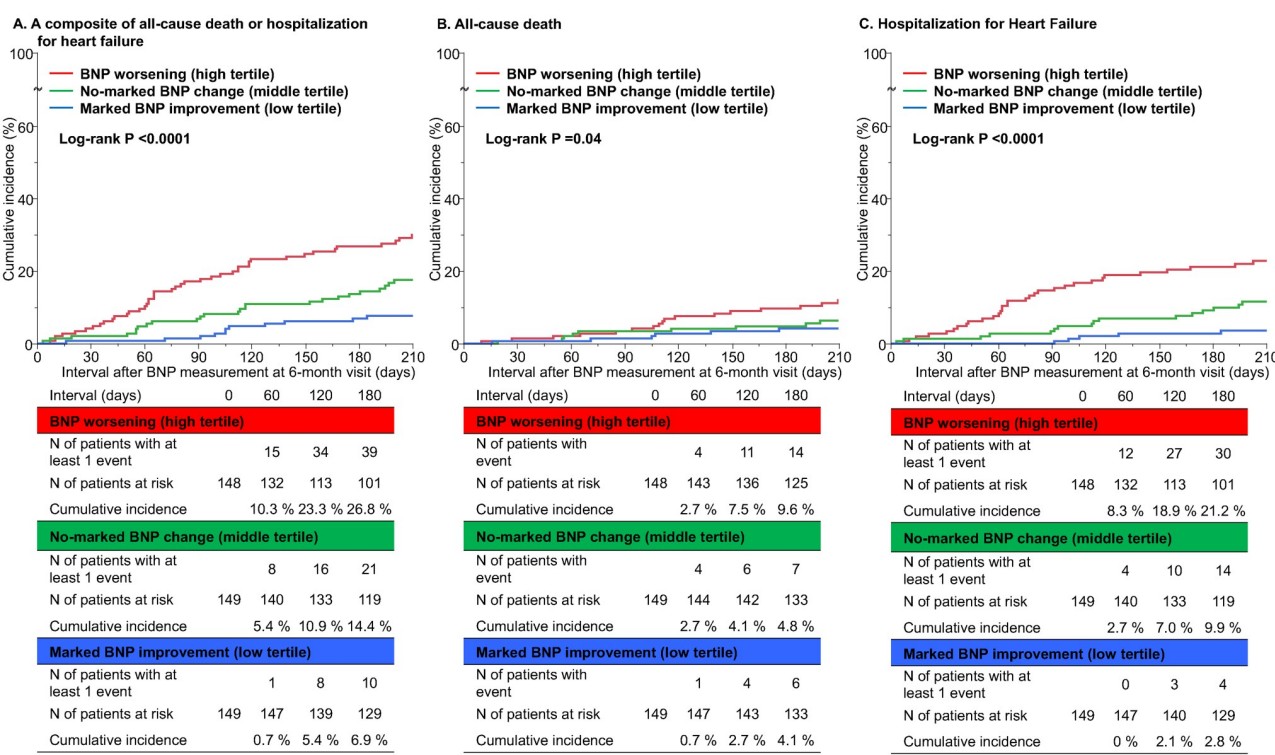

**Fig 4. Kaplan Meier curves for (A) the primary outcome measure, (B) all-cause death, and (C) hospitalization for heart failure.** The primary outcome measure was defined as a composite of all-cause death or hospitalization for heart failure. BNP, brain natriuretic peptide.

**Table 3. Clinical outcomes.**

| Clinical outcome measures | Categorized group | N of patients with event/N of patients at risk (Cumulative 180-day incidence) | Crude HR (95% CI) | P value | Adjusted HR (95% CI) | P value |
|---|---|---|---|---|---|---|
| **Primary outcome measure (a composite of all-cause death or hospitalization for heart failure)** | | | | | | |
| | BNP worsening | 39/101 (26.8%) | 4.55 (2.44–9.29) | <0.0001 | 3.43 (1.51–8.46) | 0.003 |
| | No-marked BNP change | 21/119 (14.4%) | 2.39 (1.20–5.05) | 0.01 | 1.79 (0.76–4.45) | 0.19 |
| | Marked BNP improvement | 10/129 (6.9%) | 1 (Reference) | | 1 (Reference) | |
| **All-cause death** | | | | | | |
| | BNP worsening | 14/125 (9.6%) | 2.94 (1.22–8.14) | 0.02 | 1.81 (0.49–7.30) | 0.38 |
| | No-marked BNP change | 7/133 (4.8%) | 1.51 (0.54–4.51) | 0.43 | 1.67 (0.46–6.51) | 0.43 |
| | Marked BNP improvement | 6/133 (4.1%) | 1 (Reference) | | 1 (Reference) | |
| **Hospitalization for heart failure** | | | | | | |
| | BNP worsening | 30/101 (21.2%) | 7.45 (3.17–21.8) | <0.0001 | 5.35 (1.83–19.7) | 0.001 |
| | No-marked BNP change | 14/119 (9.9%) | 3.36 (1.32–10.3) | 0.01 | 1.87 (0.58–7.22) | 0.30 |
| | Marked BNP improvement | 4/129 (2.8%) | 1 (Reference) | | 1 (Reference) | |

The Cox proportional hazards regression model was constructed adjusting for 10 clinically relevant risk-adjusting variables: age $\geq$80 years, sex, LVEF <40% by echocardiography, BNP at discharge, eGFR <30ml/min/1.73m$^2$, albumin <3.0 g/dL, diuretics, ACE-I or ARB, β-blocker and MRA.

Diuretics included loop diuretic, thiazide and tolvaptan.

LVEF, left ventricular ejection fraction; BNP, brain natriuretic peptide; eGFR, estimated glomerular filtration rate; ACE-I, angiotensin-converting enzyme inhibitor; ARB, angiotensin-receptor blocker; MRA, mineralocorticoid receptor antagonist; HR, hazard ratio; CI, confidence interval.

measure (HR: 3.43, 95%CI: 1.51–8.46, P = 0.003) and for hospitalization for HF (HR: 5.35, 95%CI: 1.83–19.7, P = 0.0001), whereas the adjusted risk of the BNP worsening group relative to the marked BNP improvement group was no longer significant for all-cause death (HR: 1.81, 95%CI: 0.49–7.30 P = 0.38) (Table 3). We showed the figures of changes in BNP during discharge and 6-month visit in each group (S2 Fig).

## Sensitivity analyses

When we evaluated age, LVEF, BNP at discharge, eGFR and albumin as a continuous variable, the excess risk of the BNP worsening group relative to the marked BNP improvement group remained significant for the primary outcome measure (HR: 3.47, 95%CI: 1.46–8.21, P = 0.005) and for hospitalization for HF (HR: 5.19, 95%CI: 1.59–17.0, P = 0.007), whereas the adjusted risk of the BNP worsening group relative to the marked BNP improvement group was not significant for all-cause death (HR: 1.94, 95%CI: 0.52–7.27 P = 0.32) (S2 Table), which was consistent with the main analysis.

## Post-hoc subgroup analyses

There were no significant interactions between the risk of the percent change in BNP for the primary outcome measure and all the subgroup factors except for the use of ACE-I or ARB (S3

Table); the magnitude of the effect of the BNP worsening group for the primary outcome measure was greater in patients with the use of ACE-I or ARB.

## Clinical outcomes in the subpopulations according to tertiles of BNP level at discharge

In all the subpopulations, the cumulative 180-day incidences of the primary outcome measure were significantly or numerically higher in the BNP worsening group and the no-marked BNP change group than in the marked BNP improvement group (S3 Fig).

## Discussion

The main findings of the present study are as follows; 1) Patients in the BNP worsening group had higher prevalence of non-HFrEF at discharge with minimal change in LAD and LVEDD from discharge to 6-month visit; 2) Percent change in BNP was associated with a subsequent risk for a composite of all-cause death or hospitalization for HF after adjustment of the absolute BNP values at discharge; 3) The direction of BNP changes from discharge to 6-month visit might be affected by regression to the mean.

There are large systemic differences among BNP levels provided by commercial immunoassay methods because of considerable chemical and structural heterogeneity of BNP circulating in human blood [12]. Franzini et al. reported that the IRMA method (by Shionogi's Diagnostic Division, Japan), the ADVIA method for the Centaur platform (by Siemens Health Care Diagnostics) and the ST-AIA-PACK method for the AIA platform (by TOSOH Corporation, Tokyo, Japan) measured greatly lower (up to the half) BNP values in comparison with other immunoassays, such as the POCT Triage method (by Alere Diagnostics), the BNP Triage Biosite for Access and UniCell DxI platforms (by Beckman Coulter Diagnostics), the MEIA method for the AxSYM platform and the chemiluminescent microparticle immunoassay for ARCHITECT platform (both by Abbotts Diagnostics) [13]. Additionally, BNP levels were affected by sex, age, heart rate, renal function and body mass index [12]. In Japan, BNP level is measured by the former immunoassay methods. The reference interval of ST-AIA-PACK method in a healthy population is ≤18.4 ng/L. The sensitivity and specificity of BNP at a threshold of ≤100 ng/L were 0.95 [95% confidence interval (CI): 0.93–0.96] and 0.63 (95% CI: 0.52 to 0.73), respectively [1].

Many previous small or large-scale studies showed that change in natriuretic peptides from hospital admission to follow-up after discharge was associated with clinical outcomes in HF patients [8, 14, 15]. Kagiyama et al. evaluated change in BNP during hospitalization for AHF as a prognostic biomarker for all-cause death [16]. It is obvious that patients with higher BNP levels than those in acute phase of HF result in unfavorable outcomes. Bettencourt et al. conducted a single-center retrospective study and numerically observed that BNP levels at AHF admission were more than 2.5 times higher than those at a stable HF condition in the AHF hospitalization group [17]. To the best of our knowledge, no previous study statistically evaluated the association of the changes in BNP from discharge to follow-up with subsequent clinical outcomes in patients with AHF.

This study showed that percent change in BNP was independently associated with the primary composite outcome measure and HF hospitalization, even after adjusting for medications for HF and BNP levels at discharge. This finding may be supported by the observation that the risk for adverse clinical events in the BNP worsening group tended to be greater in patients using ACE-I or ARB (S3 Table). On the other hand, Zhang et al. pointed out that although serial measurement of NT-proBNP is useful, the most recent value of NT-proBNP has similar predictive power [18]. The event occurred in the BNP worsening group despite of

the low BNP levels in the present study (S2 Fig). Volume expansion and pressure overload caused by worsening HF and leading to wall stress stimulate synthesis and secretion of BNP mainly from cardiac ventricular myocytes [19, 20]. Conversely, increase in BNP reflects volume expansion and pressure overload, which may attribute to clinical events. In asymptomatic HF patients, cardiac remodeling was an independent predictor of clinical events [21].

As shown in Table 1, although BNP level at discharge in the marked BNP improvement group was significantly higher, BNP level at 6-month visit was significantly lower than the other groups. This reverse association may be affected by regression to the mean and attributed to a higher prevalence of cardioprotective drugs use. Those with higher BNP level were more likely to be treated intensively; thus, if HF management was successful, they were more likely to be in the marked BNP improvement group with better final outcomes.

At 6-month visit, regardless of no difference in LVEDD and LVEF among 3 groups, the BNP worsening group had a higher LVMI indicating pressure overload and other echocardiographic findings of congestive status including a higher TRPG, greater LAD and IVC diameter and a higher prevalence of MR and TR, which indicate volume expansion. These features may be linked to the increased BNP value. With reference to echocardiographic changes from discharge to 6-month visit, the BNP worsening group relative to the other groups showed minimal improvement of echocardiographic parameters, indicating a lack of LV and LA reverse remodeling. There might be several reasons for this lack of LV and LA reverse remodeling. First, the BNP worsening group had a higher prevalence of previous myocardial infarction and atrial arrythmias at the 6-month visit. Ischemic cardiomyopathy is known to be associated with the absence of LV reverse remodeling [22]. Atrial fibrillation is associated with atrial enlargement [23]. Second, the BNP worsening group had a lower prevalence of β-blocker use at 6-month visit, which is one of the key drugs for cardiac reverse remodeling [24]. Third, the BNP worsening group had smaller LVEDD and higher LVEF at discharge, indicating that there was a possibility of little room of reverse remodeling. Further, the BNP worsening group had a numerical increase in LAD, which was considered to be the reflection of elevated end-diastolic pressure of LV.

Regardless of absolute BNP levels, the direction of BNP changes from a stable condition at discharge may indicate disease progression or successful management of HF. The more decrease in BNP levels means the more favorable outcomes in the present study. Thus, we can modify the intensity of management for congestion if we know the changes of BNP levels in each patient. Further studies are needed to research improvement of clinical outcomes in patients with HF by adjusting HF management based on change in BNP.

## Limitations

The present study has several limitations that should be addressed. First, it is possible that absent data can alter the study results (*i.e.* selection bias); the present study population only comprised 446 patients of the 4056 patients enrolled in the KCHF registry or of the 1246 patients scheduled for a 6-month follow-up. Although 99 loss to follow-up and 23 death were excluded, there was no significant difference in BNP at discharge between 122 excluded patients and 446 analyzed patients (248 [IQR, 90.7–502] versus 234 [IQR, 127–443], P = 0.93). Data on changes in BNP were not available in a substantial proportion of the cohort scheduled for a 6-month follow-up. BNP were not measured in a substantial proportion of patients who were followed by NT-proBNP. The measurements of BNP or NT-proBNP were basically dependent on the availability in each participating hospital. Nevertheless, the patients without the data on the change in BNP levels were older and less likely to be women, and had a lower prevalence of atrial arrythmias (S1 Table). These very significant selection of patients remains

a major limitation to this study. Second, data on medications at the 6-month visit were also not available in a substantial proportion of patients, although the characteristics of patients with available data on medications (N = 352) and without data on medications (N = 94) were not significantly different (S4 Table). A very advanced age of the study population might be a reason for us not to collect the detailed data in all patients, even if they were prospectively enrolled. Additionally, a proportion of those who used cardioprotective drugs was relatively low because the present population included many non-HFrEF patients. There is a possibility that missing detailed data might alter the study results and adjustment of medications might be inadequate. Third, the follow-up period was relatively short and the number of clinical events was relatively small in this study, which made it difficult to make extensive adjustment. There may be residual and unmeasured confounding factors related to outcomes. Forth, the BNP immunoassay methods were not collected and not designed to be uniformed among the 19 participating hospitals. Finally, high-sensitivity cardiac troponin was not included into the adjustment model because of many absent data on cardiac troponin, which might have a better cardiovascular risk stratification [25].

## Conclusion

Percent change in BNP was associated with a risk for a composite of all-cause death or hospitalization for HF after adjustment of the absolute BNP values, suggesting that observing the change in BNP levels, in addition to absolute BNP levels themselves, helps us to manage patient with HF.

## Supporting information

**S1 Table. Patient characteristics compared between patients with available data on change in BNP and those without data.** Categorical variables are presented as number (%), and continuous variables are presented as mean ± SD. BNP, brain natriuretic peptide; BMI, body mass index; AF, atrial fibrillation; AFL, atrial flutter; eGFR, estimated glomerular filtration rate; ACE-I, angiotensin converting-enzyme inhibitor; ARB, angiotensin-receptor blocker; MRA, mineralocorticoid receptor antagonist; SD, standard deviation. Diuretics included loop diuretic, thiazide or tolvaptan.
(PDF)

**S2 Table. Sensitivity analyses.** The Cox proportional hazards regression model was constructed adjusting for 10 clinically relevant risk-adjusting variables: age, LVEF, BNP at discharge, eGFR and albumin as a continuous variable and sex, diuretics, ACE-I or ARB, β-blocker and MRA. Diuretics included loop diuretic, thiazide and tolvaptan. LVEF, left ventricular ejection fraction; BNP, brain natriuretic peptide; eGFR, estimated glomerular filtration rate; ACE-I, angiotensin-converting enzyme inhibitor; ARB, angiotensin-receptor blocker; MRA, mineralocorticoid receptor antagonist; HR, hazard ratio; CI, confidence interval.
(PDF)

**S3 Table. Subgroup analysis for the primary outcome measure according to the tertiles of percent change in BNP.** Values are n/n (%). BNP, brain natriuretic peptide; LVEF, left ventricular ejection fraction; ACE-I, angiotensin-converting enzyme inhibitor; ARB, angiotensin-receptor blocker; HR, hazard ratio; CI, confidence interval.
(PDF)

**S4 Table. Patient characteristics compared between patients with available data on medications at 6-month visit and those without data.** Categorical variables are presented as number (%), and continuous variables are presented as mean ± SD or median (interquartile range).

BMI, body mass index; AF, atrial fibrillation; AFL, atrial flutter; ICH, intracranial hemorrhage; BP, blood pressure; HR, heart rate; BNP, brain natriuretic peptide; eGFR, estimated glomerular filtration rate; SD, standard deviation.
(PDF)

**S1 Fig. Histogram of percent change in BNP from discharge to 6-month visit.** % change in BNP, percent change in brain natriuretic peptide.
(PDF)

**S2 Fig. Changes in BNP during discharge and 6-month visit in (A) the marked BNP improvement group, (B) the no-marked BNP change group, and (C) the BNP worsening group.** This study population was classified into the 3 groups by percent change in BNP during discharge and 6-month visit; the marked BNP improvement group (≤-44%, N = 149), the no-marked BNP change group (>-44% and ≤22%, N = 149) and the BNP worsening group (>22%, N = 148). Red lines indicate patients with events. Blue lines indicate patients without events. BNP, brain natriuretic peptide; HF, heart failure.
(PDF)

**S3 Fig. Kaplan Meier curves for the primary outcome measure in three subpopulations according to BNP level at discharge.** The primary outcome measure was defined as a composite of all-cause death or hospitalization for heart failure. BNP, brain natriuretic peptide.
(PDF)

## Acknowledgments

The authors thank the members of the KCHF study, the other members of the participating centers.

## Author Contributions

**Conceptualization:** Masayuki Shiba, Hidenori Yaku, Yasutaka Inuzuka, Yodo Tamaki, Neiko Ozasa, Erika Yamamoto.

**Data curation:** Masayuki Shiba, Takao Kato, Hidenori Yaku, Yasutaka Inuzuka, Yodo Tamaki, Neiko Ozasa, Yuta Seko, Erika Yamamoto, Yusuke Yoshikawa, Takeshi Kitai, Yugo Yamashita, Moritake Iguchi, Kazuya Nagao, Yuichi Kawase, Takashi Morinaga, Mamoru Toyofuku, Yutaka Furukawa, Kenji Ando, Kazushige Kadota, Yukihito Sato, Yasuaki Nakagawa, Koichiro Kuwahara.

**Formal analysis:** Masayuki Shiba, Takeshi Morimoto.

**Funding acquisition:** Takao Kato, Neiko Ozasa, Koichiro Kuwahara.

**Investigation:** Masayuki Shiba, Takao Kato, Hidenori Yaku, Yasutaka Inuzuka, Yodo Tamaki, Neiko Ozasa, Yuta Seko, Erika Yamamoto, Yusuke Yoshikawa, Takeshi Kitai, Yugo Yamashita, Moritake Iguchi, Kazuya Nagao, Yuichi Kawase, Takashi Morinaga, Mamoru Toyofuku, Yutaka Furukawa, Kenji Ando, Kazushige Kadota, Yukihito Sato, Yasuaki Nakagawa, Koichiro Kuwahara.

**Methodology:** Masayuki Shiba, Takao Kato, Takeshi Morimoto, Hidenori Yaku, Yasutaka Inuzuka, Yodo Tamaki, Neiko Ozasa, Yuta Seko, Erika Yamamoto, Yusuke Yoshikawa, Takeshi Kitai, Yugo Yamashita, Moritake Iguchi, Kazuya Nagao, Yuichi Kawase, Takashi Morinaga, Mamoru Toyofuku, Yutaka Furukawa, Kenji Ando, Kazushige Kadota, Yukihito Sato, Yasuaki Nakagawa, Koichiro Kuwahara, Takeshi Kimura.

**Project administration:** Takao Kato, Takeshi Morimoto, Hidenori Yaku, Yasutaka Inuzuka, Yodo Tamaki, Neiko Ozasa, Erika Yamamoto, Takeshi Kimura.

**Supervision:** Takeshi Kimura.

**Writing – original draft:** Masayuki Shiba.

**Writing – review & editing:** Takao Kato, Takeshi Morimoto, Hidenori Yaku, Yasutaka Inuzuka, Yodo Tamaki, Neiko Ozasa, Yuta Seko, Erika Yamamoto, Yusuke Yoshikawa, Takeshi Kitai, Yugo Yamashita, Moritake Iguchi, Kazuya Nagao, Yuichi Kawase, Takashi Morinaga, Mamoru Toyofuku, Yutaka Furukawa, Kenji Ando, Kazushige Kadota, Yukihito Sato, Yasuaki Nakagawa, Koichiro Kuwahara, Takeshi Kimura.

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
