## [Decision Letter · Decision Letter 0]

23 Nov 2021

PONE-D-21-35110Changes in BNP levels from discharge to 6-month visit predict subsequent outcomes in patients with acute heart failurePLOS ONE

Dear Dr. Kato,

Thank you for submitting your manuscript to PLOS ONE. After careful consideration, we feel that it has merit but does not fully meet PLOS ONE’s publication criteria as it currently stands. Therefore, we invite you to submit a revised version of the manuscript that addresses the points raised during the review process.

The manuscript has been carefully evaluated by 2 external reviewers and they found the manuscript potentially of interest. However, the referees have identified some conceptual and methodological problems and they have required additional information and clarifications from the authors that need to be provided.

We look forward to receiving your revised manuscript.

Kind regards,

Claudio Passino, MD

Academic Editor

PLOS ONE

Journal Requirements:

(This work was supported by the Japan Agency for Medical Research and Development [18059186] (Drs T. Kato, Kuwahara, and Ozasa))

(This work was supported by the Japan Agency for Medical Research and Development [18059186] (T.K, K.K, and N.O).

https://www.amed.go.jp/en/

The funders had no role in study design, data collection and analysis, decision to publish, or preparation of the manuscript.)

Reviewers' comments:

Reviewer's Responses to Questions

**Comments to the Author**

1. Is the manuscript technically sound, and do the data support the conclusions?

Reviewer #1: Partly

Reviewer #2: Yes

2. Has the statistical analysis been performed appropriately and rigorously? 

Reviewer #1: Yes

Reviewer #2: Yes

3. Have the authors made all data underlying the findings in their manuscript fully available?

Reviewer #1: Yes

Reviewer #2: Yes

4. Is the manuscript presented in an intelligible fashion and written in standard English?

Reviewer #1: Yes

Reviewer #2: Yes

5. Review Comments to the Author

Reviewer #1: To the Authors

General Considerations

The aim of this study was to investigate the association between changes in BNP from discharge to 6-month visit and subsequent clinical outcomes in patients with acute HF. Authors enrolled 446 patients with available paired BNP data at discharge and 6-month index visit, classified according to the tertiles of BNP concentrations: low (≤-44 %, N=149), middle tertile (>-44 % and ≤22 %, N=149) and high (>22 %, N=148). The main results of this study were that % changes in BNP among the 3 groups of HF patients were associated with a subsequent risk for a composite of all-cause death and hospitalization for HF. Authors concluded that changes in BNP levels may help to manage patient with HF.

There is an extensive literature supporting the clinical relevance of changes of BNP levels in monitoring patients with an acute episode of HF (249 articles in PubMed using the key words: acute heart failure, BNP changes; 101 articles using the key words: acute heart failure, BNP changes, outcome). Authors should better indicate in Introduction (or Discussion) sections of the revised manuscript the originality of this article compared to other previous studies or to other more recently published (for example: Bettencourt P et al. JACEP Open 2021;2:e12448).

I have some specific points to address to the Authors in order to further improve the scientific message of this article.

Specific Points

1. BNP assay. Authors compare 3 different groups of HF patients divided according to the levels of BNP. It is well known that there are several assay methods for BNP showing very large systematic differences (up to folds) in the concentrations measured in healthy subjects and HF patients (for reviews about this issue: Clerico A et al. Clin Chim Acta 2015;443:17-24; Clerico A. et al. Future Cardiol 2016:12:573-84)). Authors should add a specific paragraph in the revised manuscript reporting the analytical characteristics and performances and reference interval values measured in a healthy population (compared to age and sex to HF patients) with the BNP method used in this study. The international guidelines by IFCC (International Federation of Clinical Chemistry) recommend that the BNP concentration should be reported as ng/L (not pg/mL) (Apple FS et al. Circulation 2007;116;e95-e98).

2. Table 1, BNP at discharge. An interesting data reported in Table 1 is that patients included in the first tertile have a BNP discharge values significantly lower than the other two tertiles. These results should be discussed more in detail in the revised manuscript. Authors should better explain this reverse association between elevated BNP at discharge and better final outcome.

3. A limitation of this study is that the measurement of cardiac troponins using high-sensitivity assays (i.e., hs-cTnI and hs-cTnT methods) was not performed in this study. Several recent studies have demonstrated that hs-cTnI and hs-cTnT may have a better risk stratification both in apparently healthy subjects and patients with cardiac disease (including HF) than natriuretic peptides (BNP or NT-proBNP) (Farmakis D et al. Eur Heart J 2020;41:4050-6; Clerico A et al. Clin Chem Lab Med 2020;59:79-90; Harrison N et al. Curr Heart Fail Rep 2019;16:21-31; Rosa GM et al. Eur J Clin Invest 2019;49:e13044; Perna ER Minerva Carioangiol 2016;64:165-80; Aimo A et al. Circulation 2018;137:286.297; Aimo A et al. Int J Cardiol 2019;277:166-172). This important point should be discussed by the Authors.

Reviewer #2: In the present paper, Shiba and colleagues aim to investigate "the association between the changes in BNP levels and subsequent clinical outcomes in patients with AHF". The Authors collect data from the Kyoto Congestive Heart Failure (KCHF) registry (n=446) whe received BNP testing at discharge after an AHF episode and at 6 month. They report that the incidence of the primary end-point (a composite of all-cause death or hospitalization for HF) was significantly higher in patients with stable/increased BNP vs those with decreasing BNP levels.

The message of the paper is rather clear, and the conclusions are supported by the provided evidence. Still, some major points, as listed below, need to be addressed.

Major points

- How did the Authors select the variables to be included in the multivariable model? Why did they dichotomized variables such as LVEF or eGFR?

- Further to the pre-specified subgroups, a subset analysis should have been performed in patients with/without atrial fibrillation

- How/when was exactly defined atrial fibrillation? How did the Authors account for eventual changes in background rhythm

- The cause for AHF should be reported in the study population. Moreover, whether it was a de novo vs an acutely decompensated HF should be also mentioned.

Minor points

- In the methods section, end-stage renal disease was defined according to eGFR, while exsclusion criteria were based on serum creatinine. Please clarify.

- The number of events in the whole population and in subgroups according to BNP changes should be clearly reported in the Methods section.

6. PLOS authors have the option to publish the peer review history of their article (what does this mean?). If published, this will include your full peer review and any attached files.

Reviewer #1: No

Reviewer #2: No

---

## [Author Response · Author response to Decision Letter 0]

12 Dec 2021

Response

We appreciate the information about your journal format and the editor`s and reviewers` careful evaluation and suggestion. We have replayed to the reviewers’ comments and revised our manuscript.

We have modified font size of heading, figure captions, acknowledgements (Page 33, line 9-11) and supporting information captions (Page 40, line 1-Page 42, line 15). We have removed any funding-related text from the manuscript. We have added the section of Data Availability Statement: All relevant data are within the manuscript and its Supporting Information files (Page 10, line 15-16).

Reviewers' comments:

Reviewer's Responses to Questions

Reviewer #1: To the Authors

General Considerations

The aim of this study was to investigate the association between changes in BNP from discharge to 6-month visit and subsequent clinical outcomes in patients with acute HF. Authors enrolled 446 patients with available paired BNP data at discharge and 6-month index visit, classified according to the tertiles of

BNP concentrations: low (≤-44 %, N=149), middle tertile (>-44 % and ≤22 %, N=149) and high (>22 %, N=148). The main results of this study were that % changes in BNP among the 3 groups of HF patients were associated with a subsequent risk for a composite of all-cause death and hospitalization for HF. Authors concluded that changes in BNP levels may help to manage patient with HF.

There is an extensive literature supporting the clinical relevance of changes of BNP levels in monitoring patients with an acute episode of HF (249 articles in PubMed using the key words: acute heart failure, BNP changes; 101 articles using the key words: acute heart failure, BNP changes, outcome). Authors

should better indicate in Introduction (or Discussion) sections of the revised manuscript the originality of this article compared to other previous studies or to other more recently published (for example: Bettencourt P et al. JACEP Open 2021;2:e12448).

I have some specific points to address to the Authors in order to further improve the scientific message of this article. 

Response

We thank the reviewer for the careful assessment, the appropriate suggestion and showing many specific references. We have described the originality of this article compared to the previous study (Bettencourt P et al. JACEP Open.2021;2:e12448) in the section of Discussion. Bettencourt`s concept and purpose was the same as ours. Bettencourt et al. conducted a single-center retrospective study and numerically observed that BNP levels at AHF admission were more than 2.5 times higher than those at a stable HF condition in the AHF hospitalization group. However, he didn`t statistically evaluate the relationship between change in BNP from a stable condition and the risk for clinical outcomes. This is our advantage. (Page 28, line 11-15).

Specific Points

1. BNP assay. Authors compare 3 different groups of HF patients divided according to the levels of BNP. It is well known that there are several assay methods for BNP showing very large systematic differences (up to folds) in the concentrations measured in healthy subjects and HF patients (for reviews about this issue: Clerico A et al. Clin Chim Acta 2015;443:17-24; Clerico A. et al. Future Cardiol 2016:12:573-84)). Authors should add a specific paragraph in the revised manuscript reporting the analytical characteristics and performances and reference interval values measured in a healthy population (compared to age and sex to HF patients) with the BNP method used in this study. The international guidelines by IFCC (International Federation of Clinical Chemistry) recommend that the BNP concentration should be reported as ng/L (not pg/mL) (Apple FS et al. Circulation 2007;116;e95-e98).

Response

We appreciate your comment. We have change from pg/mL to ng/L along with your recommendation. We have added the paragraph about BNP Assay in the section of Discussion and reported the analytical characteristics and performances and reference interval values measured in a healthy population (Page 27, line 8-Page 28, line 5). The BNP immunoassay methods were not collected and not designed to be uniformed among the 19 participating hospitals. We have added this into the section of Limitation (Page 32, line 15-17). 

2. Table 1, BNP at discharge. An interesting data reported in Table 1 is that patients included in the first tertile have a BNP discharge values significantly lower than the other two tertiles. These results should be discussed more in detail in the revised manuscript. Authors should better explain this reverse association between elevated BNP at discharge and better final outcome.

Response

We agree on your great insight. We think that this reverse association may be affected by regression to the mean and attributed to a higher prevalence of cardioprotective drugs use (Page 29, line 11-17). We had added the following discussions:

In the section of Discussion

As shown in Table 1, although BNP level at discharge in the marked BNP improvement group was significantly higher, BNP level at 6-month visit was significantly lower than the other groups. This reverse association may be affected by regression to the mean and attributed to a higher prevalence of cardioprotective drugs use. Those with higher BNP levels were more likely to be treated intensively; thus, if HF management was successful, they were more likely to be in the marked BNP improvement group with better final outcomes.

3. A limitation of this study is that the measurement of cardiac troponins using high-sensitivity assays (i.e., hs-cTnI and hs-cTnT methods) was not performed in this study. Several recent studies have demonstrated that hs-cTnI and hs-cTnT may have a better risk stratification both in apparently healthy subjects and patients with cardiac disease (including HF) than natriuretic peptides (BNP or NT-proBNP) (Farmakis D et al. Eur Heart J 2020;41:4050-6; Clerico A et al. Clin Chem Lab Med 2020;59:79-90; Harrison N et al. Curr Heart Fail Rep 2019;16:21-31; Rosa GM et al. Eur J Clin Invest 2019;49:e13044; Perna ER Minerva Carioangiol 2016;64:165-80; Aimo A et al. Circulation 2018;137:286.297; Aimo A et al. Int J Cardiol 2019;277:166-172). This important point should be discussed by the Authors.

Response

We thank the reviewer for the valuable suggestion. We tried to include cardiac troponin into the adjustment model. However, we could not include because of many absent data on cardiac troponin. We have discussed this in the section of Limitation (Page 32, line 17-Page 33, line 1).

In the section of Limitation

Finally, high-sensitivity cardiac troponin was not included into the adjustment model because of many absent data on cardiac troponin, which might have a better cardiovascular risk stratification (25).

Reviewer #2: In the present paper, Shiba and colleagues aim to investigate "the association between the changes in BNP levels and subsequent clinical outcomes in patients with AHF". The Authors collect data from the Kyoto Congestive Heart Failure (KCHF) registry (n=446) who received BNP testing at discharge after an AHF episode and at 6 month. They report that the incidence of the primary end-point (a composite of all-cause death or hospitalization for HF) was significantly higher in patients with stable/increased BNP vs those with decreasing BNP levels.

The message of the paper is rather clear, and the conclusions are supported by the provided evidence. Still, some major points, as listed below, need to be addressed.

Response

We appreciate the positive comments, the careful assessment, and the clear suggestions.

Major points

- How did the Authors select the variables to be included in the multivariable model? Why did they dichotomized variables such as LVEF or eGFR? 

Response:

We thank the reviewer for your comments. From the adjusting variables which was preliminarily designed in our previous studies, we selected 10 adjusting factors more closely-related with heart failure outcomes because a few clinical events occurred in this study population. 

The use of dichotomization of continuous variables was almost consistent across our previous studies. We have added a sensitivity analysis with age, LVEF, BNP at discharge, eGFR and albumin as a continuous variable. The results were still consistent with the main analysis (S4 Table).

In the section of Materials and methods (Page 9, line 16-18)

As a sensitivity analysis, we included age, LVEF, BNP at discharge, eGFR and albumin as a continuous variable in the adjusted model in patients with available data.

In the section of Results (Page 25, line 11-18)

When we evaluated age, LVEF, BNP at discharge, eGFR and albumin as a continuous variable, the excess risk of the BNP worsening group relative to the marked BNP improvement group remained significant for the primary outcome measure (HR: 3.47, 95%CI: 1.46-8.21, P=0.005) and for hospitalization for HF (HR: 5.19, 95%CI: 1.59-17.0, P=0.007), whereas the adjusted risk of the BNP worsening group relative to the marked BNP improvement group was not significant for all-cause death (HR: 1.94, 95%CI: 0.52-7.27 P=0.32) (S4 Table), which was consistent with the main analysis.

- Further to the pre-specified subgroups, a subset analysis should have been performed in patients with/without atrial fibrillation

- How/when was exactly defined atrial fibrillation? How did the Authors account for eventual changes in background rhythm.

Response

We thank the reviewer for your comments. We have added the definition of atrial arrythmias. Atrial arrythmias including atrial fibrillation and flutter were counted base on medical history and their events during index hospitalization and electrocardiography at 6-month visit (Page 8, line 13-15). We additionally have evaluated atrial arrythmias in the post-hoc subgroup analysis (Page 10, line 3). There was no significant interaction between the risk of the percent change in BNP for the primary outcome measure and atrial arrythmias (S5 Table).

- The cause for AHF should be reported in the study population. Moreover, whether it was a de novo vs an acutely decompensated HF should be also

mentioned.

Response

We thank the reviewer for your comments. We have added the definition and etiology of AHF. AHF is defined as de novo HF or worsening signs and symptoms of HF (11) (Page 8, line 6). Compared with the marked BNP improvement group, the BNP worsening and no-marked BNP change groups had a lower prevalence of cardiomyopathy etiology (Table 1) (Page 11, line 11-14).

Minor points

- In the methods section, end-stage renal disease was defined according to eGFR, while exsclusion criteria were based on serum creatinine. Please clarify.

Response

We appreciate specific points and apologize for a double-standard definition of end-stage renal disease. We have added the event number of the primary endpoint in the Results (Page 21, line 8-11). We defined s-Cr >3.0 mg/dL or on hemodialysis as end-stage renal disease. eGFR <30ml/min/1.73m2 was included into the adjusting model along with our previous studies.

- The number of events in the whole population and in subgroups according to BNP changes should be clearly reported in the Methods section.

Response

We appreciate your comments. We have added the number of events in the Result.

In the section of Results

During the 180-day follow-up, 39 patients in the marked BNP improvement group, 21 patients in the no-marked BNP change group and 10 patients in the BNP worsening group encountered all-cause death or hospitalization for HF (Fig 4A and Table 3) (Page 21, line 8-11).

---

## [Decision Letter · Decision Letter 1]

13 Jan 2022

Changes in BNP levels from discharge to 6-month visit predict subsequent outcomes in patients with acute heart failure

PONE-D-21-35110R1

Dear Dr. Kato,

We’re pleased to inform you that your manuscript has been judged scientifically suitable for publication and will be formally accepted for publication once it meets all outstanding technical requirements.

Kind regards,

Claudio Passino, MD

Academic Editor

PLOS ONE

Additional Editor Comments (optional):

Reviewers' comments:

Reviewer's Responses to Questions

**Comments to the Author**

1. If the authors have adequately addressed your comments raised in a previous round of review and you feel that this manuscript is now acceptable for publication, you may indicate that here to bypass the “Comments to the Author” section, enter your conflict of interest statement in the “Confidential to Editor” section, and submit your "Accept" recommendation.

Reviewer #1: All comments have been addressed

Reviewer #2: All comments have been addressed

2. Is the manuscript technically sound, and do the data support the conclusions?

Reviewer #1: Yes

Reviewer #2: Yes

3. Has the statistical analysis been performed appropriately and rigorously? 

Reviewer #1: Yes

Reviewer #2: Yes

4. Have the authors made all data underlying the findings in their manuscript fully available?

Reviewer #1: Yes

Reviewer #2: Yes

5. Is the manuscript presented in an intelligible fashion and written in standard English?

Reviewer #1: Yes

Reviewer #2: Yes

6. Review Comments to the Author

Reviewer #1: Authors revised the manuscript in accordance of suggestions made by the two Reviewers. The scientific message of the article is significantly improved now.

Reviewer #2: The Authors have properly addressed all the comments raised in the previous revision, and the manuscript has now significantly improved.

7. PLOS authors have the option to publish the peer review history of their article (what does this mean?). If published, this will include your full peer review and any attached files.

Reviewer #1: No

Reviewer #2: No

---

## [Editor Report · Acceptance letter]

20 Jan 2022

PONE-D-21-35110R1 

Changes in BNP levels from discharge to 6-month visit predict subsequent outcomes in patients with acute heart failure 

Dear Dr. Kato:

I'm pleased to inform you that your manuscript has been deemed suitable for publication in PLOS ONE. Congratulations! Your manuscript is now with our production department. 

Kind regards, 

on behalf of

Prof. Claudio Passino 

Academic Editor

PLOS ONE